# Low-dose sodium-glucose cotransporter 2 inhibitor ameliorates ischemic brain injury in mice through pericyte protection without glucose-lowering effects

Masamitsu Takashima[1], Kuniyuki Nakamura [1✉], Takuya Kiyohara[1], Yoshinobu Wakisaka [1], Masaoki Hidaka[1], Hayato Takaki[1], Kei Yamanaka[1], Tomoya Shibahara[1], Masanori Wakisaka[2], Tetsuro Ago[1] & Takanari Kitazono[1]

Antidiabetic sodium-glucose cotransporter 2 (SGLT2) inhibitors have attracted attention for their cardiorenal-protective properties beyond their glucose-lowering effect. However, their benefits in ischemic stroke remain controversial. Here we show the effects of luseogliflozin, a selective SGLT2 inhibitor, in acute ischemic stroke, using a permanent middle cerebral artery occlusion (pMCAO) model in non-diabetic mice. Pretreatment with low-dose luseogliflozin, which does not affect blood glucose levels, significantly attenuated infarct volume, blood-brain barrier disruption, and motor dysfunction after pMCAO. SGLT2 was expressed predominantly in brain pericytes and was upregulated in peri- and intra-infarct areas. Notably, luseogliflozin pretreatment reduced pericyte loss in ischemic areas. In cultured pericytes, luseogliflozin activated AMP-activated protein kinase α and increased mitochondrial transcription factor A expression and number of mitochondria, conferring resistance to oxygen-glucose deprivation. Collectively, pre-stroke inhibition of SGLT2 induces ischemic tolerance in brain pericytes independent of the glucose-lowering effect, contributing to the attenuation of ischemic brain injury.

[1] Department of Medicine and Clinical Science, Graduate School of Medical Sciences, Kyushu University, 3-1-1 Maidashi, Higashi-ku, Fukuoka 812-8582, Japan. [2] Wakisaka Internal Medicine Clinic, 1-24-19 Fujisaki, Sawara-ku, Fukuoka 814-0013, Japan. ✉email: nakamura.kuniyuki.524@m.kyushu-u.ac.jp

schemic stroke is one of the leading causes of death and dis-
ability worldwide. Diabetes mellitus is not only a primary risk
factor for stroke development but also an exacerbating factor
that contributes to poor functional outcomes[1]. Recently, sodium-
glucose cotransporter 2 (SGLT2) inhibitors, which increase glu-
cose excretion into the renal tubules and decrease blood glucose
levels, are increasingly used for patients with diabetes[2]. In addi-
tion to lowering blood glucose levels, SGLT2 inhibitors have been
shown to have various organ-protective effects. For instance,
several recent clinical studies have revealed that SGLT2 inhibitors
have beneficial effects on heart failure, albuminuria, and pro-
gression of kidney disease independent of glycemic control[3–5].
Thus, SGLT2 inhibition has attracted attention as a novel ther-
apeutic strategy for diabetes. However, in contrast to their ben-
eficial effects on cardiovascular diseases, there are still concerns
about the potential risks of SGLT2 inhibitors in the development
of ischemic stroke due to their diuretic effect. Although a recent
meta-analysis demonstrated that SGLT2 inhibitors had a neutral
effect on the risk of stroke development[6], their benefits on
functional or neurological outcomes after stroke have not yet
been proven. Furthermore, an approach for exploring direct
pharmacological action in the pathophysiology of ischemic stroke
using pretreatment of SGLT2 inhibitors is lacking. Thus, it
remains controversial whether SGLT2 inhibitors are beneficial or
detrimental to the outcomes after ischemic stroke in both
experimental and clinical studies.

Restoration of the tubuloglomerular feedback system has been
suggested as the mechanism of the renoprotective effects of
SGLT2 inhibitors, by which they reduce sodium reabsorption and
cause afferent arteriole vasoconstriction, thus preventing hyper-
filtration in diabetic nephropathy[2]. However, this hypothesis faces
several theoretical problems[7], including the compensatory upre-
gulation of SGLT1[8] and the renoprotective effect even in non-
diabetic patients[9]. In contrast, several recent studies have
demonstrated alternative mechanisms for SGLT2 inhibitor-
mediated protective effects beyond the glucose-lowering effect,
including a reduction in oxygen consumption in diabetic kidney
hypertrophy[10], inhibition of the transforming growth factor β
(TGF-β) pathway in cardiac fibrosis[11], and inhibition of inter-
leukin 6, TGF-β, and platelet-derived growth factor (PDGF) B
expression in hepatic fibrosis[12]. In addition, an SGLT2 inhibitor
attenuates renal ischemia-reperfusion injury by increasing
hypoxia-inducible factor 1α (HIF-1α)[13] or by a vascular endo-
thelial growth factor-dependent pathway[14]. These studies may
indicate the direct action of the inhibitors on SGLT2-expressing
cells; however, the expression of SGLT2 in each organ, except for
the renal proximal tubules, has not yet been fully verified.
Moreover, because most of these studies were conducted using
diabetic rodents or high doses of SGLT2 inhibitors, the effect on
blood glucose levels was not negligible. Therefore, the direct and
glycemic control-independent effects of SGLT2 inhibitors should
be determined using non-diabetic experimental models and an
extremely low dose of SGLT2 inhibitors that do not decrease
blood glucose levels but sufficiently inhibit sodium-dependent
glucose uptake in SGLT2-expressing cells.

In the present study, we aimed to investigate the pretreatment
effect of very low-dose SGLT2 inhibitor on acute ischemic stroke
in non-diabetic mice using a permanent middle cerebral artery
occlusion (pMCAO) model, with a particular focus on the blood-
brain barrier (BBB) integrity. We further explored the mechanism
of action of the inhibitor on ischemic injury, including the
expression and function of SGLT2 in the brain. We found that
SGLT2 was expressed by brain pericytes and was upregulated in
peri- and intra-infarct areas. Furthermore, administration of an
SGLT2 inhibitor attenuates pericytes loss and BBB disruption by

acquiring ischemic tolerance through activating AMP-activated
protein kinase α (AMPK) and mitochondrial biogenesis.

## Results

**Luseogliflozin administration attenuates ischemic injury after
pMCAO.** To investigate the in vivo pretreatment effect of a
selective SGLT2 inhibitor luseogliflozin on the acute phase of
ischemic stroke, non-diabetic adult male C57BL/6JJcl mice were
treated with or without 0.0001% luseogliflozin mixed with a
regular diet (equivalent to 0.1 mg/kg/day) for 2 weeks and then
subjected to pMCAO (Fig. 1a). We confirmed that low-dose
luseogliflozin did not affect plasma glucose levels at 3 days after
pMCAO as well as at baseline, as shown in the intraperitoneal

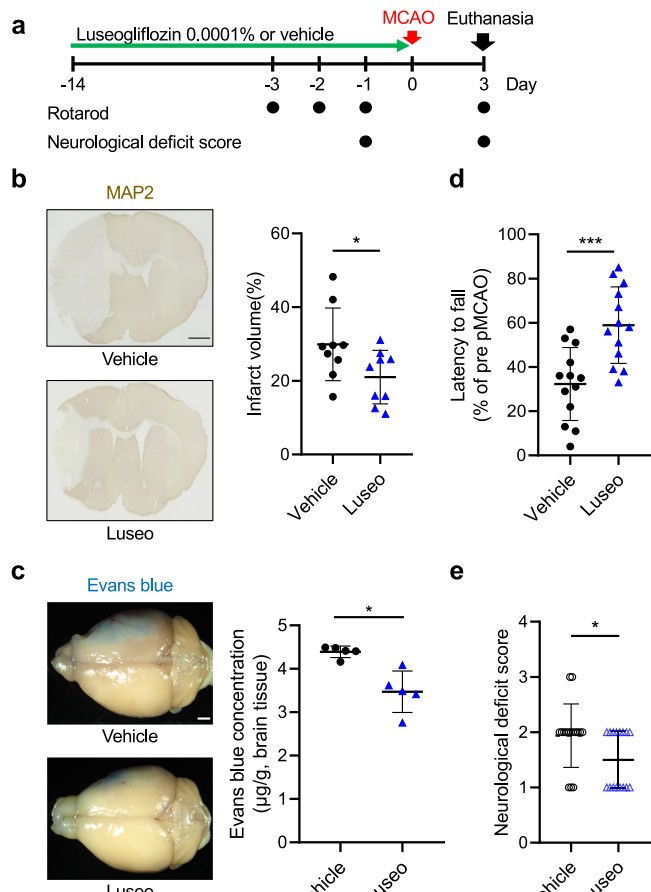

**Fig. 1 Luseogliflozin administration attenuates ischemic injury and blood-
brain barrier disruption after pMCAO. a** Luseogliflozin (Luseo), mixed in
food at a concentration of 0.0001%, was orally administered to C57BL/6JJcl
mice for 2 weeks. The mice were then subjected to pMCAO and euthanized
3 days later. Neurological assessments were performed at indicated time
points. **b** Infarct volume at 3 days after pMCAO in vehicle- and Luseo-
treated mice. Representative images of the immunohistochemistry for
MAP2 and the corresponding quantification are shown (scale bar = 1 mm,
$n = 9$ mice). **c** Evans blue dye leakage at 3 days after pMCAO in vehicle- and
Luseo-treated mice. Representative macroscopic images and the
quantification are shown (scale bar = 1 mm, $n = 5$). **d, e** Neurological
function was assessed using (**d**) rotarod test ($n = 13$ mice) and **e** modified
neurological deficit score ($n = 16$ mice) at 3 days after pMCAO in vehicle-
and Luseo-treated mice. Data are presented as dot-plots of individual
experiments and mean values ± SD. *$P < 0.05$, ***$P < 0.001$ by unpaired
$t$-test. Luseo luseogliflozin, pMCAO permanent middle cerebral artery
occlusion.

glucose tolerance test, nor blood pressure, pulse rate, or body weight. However, casual urinary glucose levels were higher in the group treated with low-dose luseogliflozin than in the vehicle-treated group (Supplementary Fig. 1a, Supplementary Tables 1 and 2).

Infarct volume was significantly lower in luseogliflozin-treated mice than in vehicle-treated mice 3 days after pMCAO, as assessed by MAP2 staining (Fig. 1b). There was no significant difference in cerebral blood flow (CBF) in ischemic lesions between vehicle- and luseogliflozin-treated mice, as evaluated by high-resolution 2D laser speckle flowmetry (Supplementary Fig. 1b). To assess BBB disruption in ischemic hemispheres, we injected an exogenous tracer, Evans blue dye, 3 days after pMCAO. Luseogliflozin pretreatment significantly reduced pMCAO-evoked Evans blue dye extravasation compared with the vehicle (Fig. 1c).

We further examined the effect of luseogliflozin on mouse behavior after pMCAO. At baseline, there were no significant differences in the latencies before falling off the accelerated rotarod between vehicle- and luseogliflozin-treated mice (Supplementary Fig. 1c). In parallel with infarct volume, neurological function was better in the luseogliflozin-treated mice 3 days after pMCAO than in the vehicle-treated group, as assessed by fall latency time (Fig. 1d) and the modified neurological deficit score (Fig. 1e).

These results indicate that administration of a low-dose SGLT2 inhibitor prior to ischemic stroke attenuates BBB disruption, resulting in lower infarct volume and attenuation of neurological symptoms after ischemic stroke.

**SGLT2 is expressed in pericytes, and luseogliflozin prevents pericyte loss after pMCAO.** To explore the expression and function of SGLT2 in the brain, we evaluated the spatial and temporal changes in SGLT2 expression after pMCAO using quantitative polymerase chain reaction (PCR) and immuno-fluorescence with an anti-SGLT2 antibody, whose specificity was confirmed by its corresponding human SGLT2 peptide and mice kidney sections as the positive control (Supplementary Fig. 2a–b). SGLT2 expression was gradually increased in the peri-infarct areas and then in the intra-infarct areas from 7 to 14 days after pMCAO (Fig. 2a–c). The upregulated SGLT2 was co-localized with CD13 or PDGFRβ, the markers for pericytes, at the abluminal side of CD31-positive vascular endothelial cells 7 days after pMCAO (Fig. 2b–c, Supplementary Fig. 2c–d). We further confirmed that brain microvessels isolated from adult mice expressed SGLT2, which was co-localized with CD13 (Fig. 2d).

We then examined the effects of luseogliflozin pretreatment on pericyte coverage in ischemic areas after pMCAO. Double immunofluorescence analysis revealed that the coverage of CD13-positive pericytes around CD31-positive endothelial cells within the infarct area was significantly higher in the luseogli-flozin group than in the vehicle group 3 days after pMCAO (Fig. 2e). These results indicate that brain pericytes express SGLT2 and that luseogliflozin prevents pericyte loss around endothelial cells after acute ischemic stroke.

**Glucose uptake by brain pericytes via SGLT2.** We used reverse transcription-polymerase chain reaction (RT-PCR) to verify that the expression of *SLC5A2* (SGLT2) in cultured human brain microvascular pericytes was comparable to that in HEK-293T cells and HK2 cells (Fig. 3a). Cultured pericytes also expressed *SLC2A1* (glucose transporter [GLUT] 1), *SLC2A3* (GLUT3), *SLC2A4* (GLUT4), and *SLC5A1* (SGLT1) (Supplementary Fig. 3a). Then, we performed immunoblotting to examine the SGLT2 expression at protein levels. To confirm the specificity of

anti-SGLT2 antibody in advance, we used HEK-293T cells with *SLC5A2* overexpression and found increased double bands around 55–60 kDa, which had almost the same size as FLAG-tag, indicating overexpressed SGLT2 protein (Supplementary Fig. 3b). Although the total cell lysates of cultured brain pericytes showed weak bands around 55–60 kDa for SGLT2, their membrane fractions obtained by ultracentrifugation demonstrated the defi-nite expression of SGLT2 as well as cell surface protein CD13 (Fig. 3b). A membrane protein, SGLT2, is highly glycosylated in vivo. The ~60-kDa bands for SGLT2 in brain pericytes shifted to ~50 kDa after treatment with PNGase F, indicating protein glycosylation (Fig. 3c). Immunofluorescence imaging of cultured pericytes further demonstrated that SGLT2 was expressed by pericytes, including the cell surface, and was co-localized with CD13 (Fig. 3d).

To assess whether pericyte SGLT2 was functional, we performed a glucose uptake assay using 2-deoxy-2-[(7-nitro-2,1,3-benzoxa-diazol-4-yl) amino]-D-glucose (2-NBDG), a fluorescent-labeled deoxyglucose analog. Treatment with luseogliflozin inhibited sodium-dependent glucose uptake in a dose-dependent manner. The uptake of 2-NBDG in $Na^+$-containing buffered saline with 1000 nmol/L luseogliflozin was low and was similar to that in $Na^+$-free buffered saline in cultured pericytes (Fig. 3e–f).

**SGLT2 in pericytes is upregulated by oxygen-glucose deprivation.** To explore the mechanism of SGLT2 gene regulation, we studied SGLT2 expression in treated cultured brain pericytes under various conditions. Oxygen-glucose deprivation (OGD) at 0 mmol/L glucose and 1% $O_2$ for 16 h increased the expression of SGLT2 in pericytes at mRNA level (Fig. 4a). Furthermore, immunofluorescence imagings revealed that its expression was increased at the cell surface under OGD, indicating the trafficking from intracellular organelles to cytoplasmic membrane (Fig. 4b). The upregulated *SLC5A2* expression returned to baseline levels after treatment with normal glucose (5 mmol/L) and oxygen (20%) concentrations. In contrast, *SLC5A2* mRNA levels were not significantly altered under high glucose level conditions (Sup-plementary Fig. 4). These results indicate that SGLT2 expression in brain pericytes may be upregulated during OGD conditions that mimic ischemia.

**Pretreatment with luseogliflozin reduces pericyte death under OGD through AMPKα activation and mitochondrial bio-synthesis.** To clarify how luseogliflozin attenuated pericyte loss during ischemic brain injury, we focused on its effects on AMPKα. Treatment with luseogliflozin for 24 h induced the phosphorylation of AMPKα, that is, its activation, in a dose-dependent manner in cultured pericytes (Fig. 5a).

AMPKα activation enhances mitochondrial biogenesis via proliferator-activated receptor gamma co-activator 1α (PGC-1α) and nuclear respiratory factor (NRF) 1 and 2 signaling[15,16]. Treatment with luseogliflozin for 24 h significantly upregulated the expression of PGC-1α, NRF1, NRF2, and mitochondrial transcription factor A (TFAM), which regulates mitochondrial function, in cultured pericytes in a dose-dependent manner (Fig. 5a–b, Supplementary Fig. 5a–b). Subsequently, treatment with 1000 nmol/L of luseogliflozin for 24 h significantly increased the mitochondrial DNA (mtDNA)/nuclear DNA (nDNA) ratio (Supplementary Fig. 5c). We also quantified the mitochondrial number using MitoTracker, a permeable selective probe for high-potential mitochondria. The results showed that treatment with 100 and 1000 nmol/L luseogliflozin for 24 h significantly increased the number of mitochondria in pericytes (Fig. 5c). Therefore, SGLT2 inhibition may reduce intracellular glucose and ATP levels and promote mitochondrial biosynthesis via AMPKα

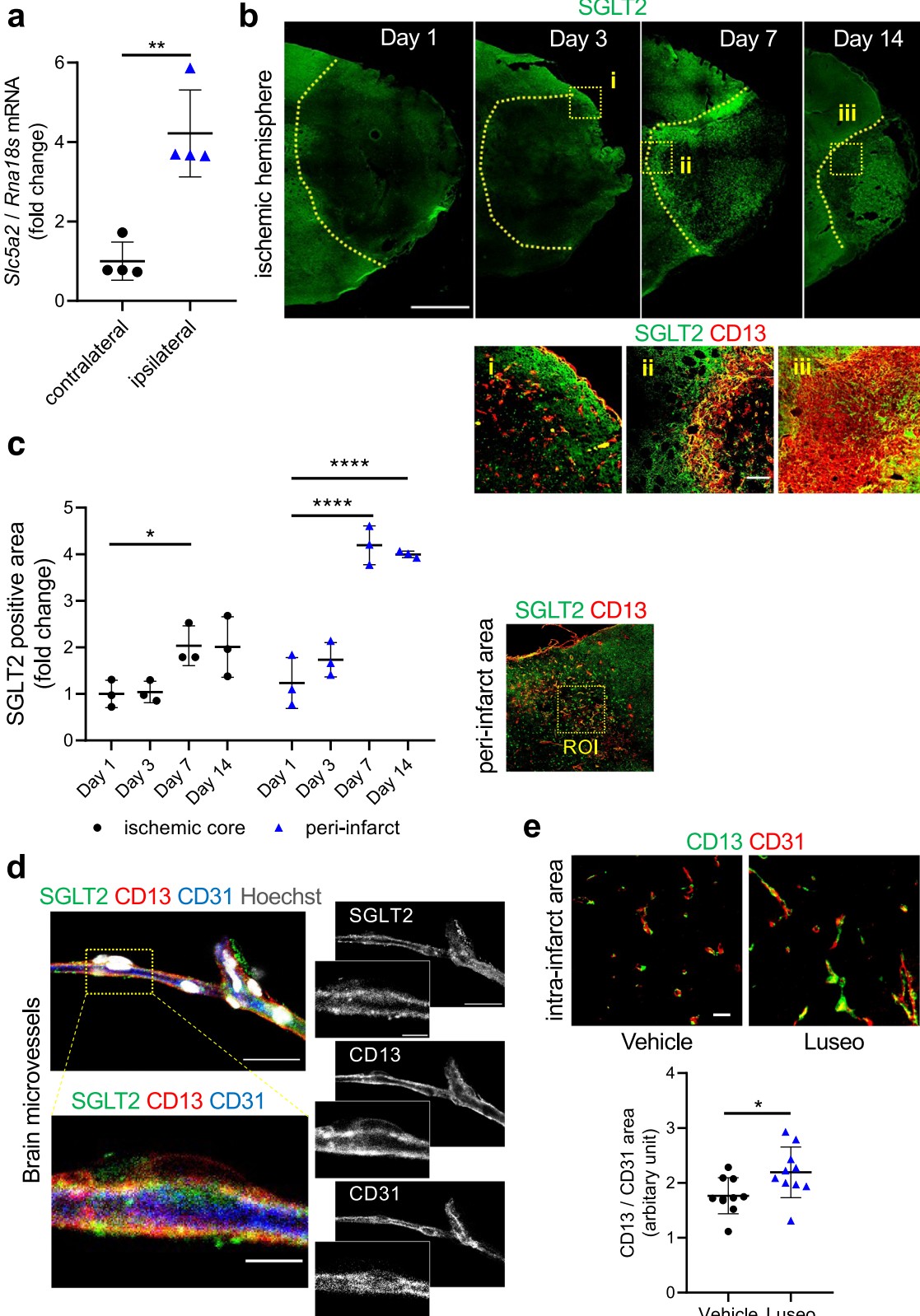

activation in brain pericytes. Consistently, pretreatment with 1000 nmol/L luseogliflozin for 24 h significantly attenuated OGD-induced cell death in cultured pericytes (Fig. 5d), indicating the acquisition of ischemic tolerance in brain pericytes.

Finally, we examined AMPKα activation in vivo using the homogenized cerebral cortex of mice treated with luseogliflozin for 14 days. AMPKα was strongly phosphorylated in the luseogliflozin group (Fig. 5e), indicating AMPKα activation in response to treatment with luseogliflozin in the mouse brain.

## Discussion

We investigated the pretreatment effects of low-dose luseogliflozin, an SGLT2 inhibitor, in a mouse model of ischemic stroke

**Fig. 2 SGLT2 is expressed in brain pericytes, and luseogliflozin prevents pericyte loss after pMCAO. a–c** Adult C57BL/6JJcl mice were subjected to pMCAO. **a** The expression level of *Slc5a2* mRNA (SGLT2) was quantified using qPCR in ischemic lesions and in the contralateral cortex at 7 days after pMCAO ($n = 4$ mice). **b** Representative images of immunofluorescence for SGLT2 at 1, 3, 7, and 14 days after pMCAO (upper panels, scale bar = 1 mm). Yellow dotted line indicates infarct area. Representative magnified images of the dotted squares in the upper panels (**i-iii**) in peri- or intra-infarct areas, which were labeled with SGLT2 (green) and CD13 (red) (scale bar = 100 μm), are shown in the lower panels. (**c**) The right panel shows a representative image of immunofluorescence for SGLT2 (green) and CD13 (red) in the peri-infarct area at 7 days after pMCAO. Yellow dotted line indicates a ROI for the subsequent quantification. SGLT2-positive areas were quantified in peri- and intrainfarct areas at 1, 3, 7, 14 days after pMCAO and are shown as fold change to day 1 expression (right panel, $n = 3$ mice). **d** Brain microvessels were isolated from healthy C57BL/6JJcl mice and labeled with SGLT2 (green), CD13 (red), CD31 (blue), and Hoechst (gray) (upper left panel, scale bar = 20 μm). The lower left panel and insets of right panels indicate a higher magnification of the boxed region (scale bar = 5 μm). **e** Representative images of immunofluorescence for CD13 (green) and CD31 (red) in intra-infarct areas and the quantification of pericyte coverage (CD13 / CD31) at 3 days after pMCAO in vehicle- and luseogliflozin-treated mice (scale bar = 20 μm, $n = 10$ mice). Data are presented as dot-plots of individual experiments and mean values ± SD. *$P < 0.05$, **$P < 0.01$, ****$P < 0.0001$ by unpaired *t*-test (**a**, **e**) or by One-way ANOVA followed by Bonferroni's post hoc test (**c**). SGLT2 sodium-glucose cotransporter 2, Luseo luseogliflozin, pMCAO permanent middle cerebral artery occlusion, ROI region of interest.

and assessed the expression and function of SGLT2 in the brain. First, we found that administration of low-dose luseogliflozin prior to pMCAO reduced infarct size and attenuated BBB breakdown. Second, SGLT2 was expressed predominantly in the pericytes in the brain and upregulated in peri- and intra-infarct areas after pMCAO. Moreover, pre-stroke administration of luseogliflozin attenuated pericyte loss around endothelial cells in the ischemic area. Third, treatment with luseogliflozin activated the AMPKα-PGC-1α pathway and promoted mitochondrial biogenesis in cultured pericytes, conferring ischemic tolerance (Fig. 6). This report demonstrates SGLT2 expression in brain pericytes and the potential role of SGLT2 inhibitors in BBB maintenance in acute ischemic stroke without the glucose-lowering effect.

The most pivotal point of our study design is the examination of the pretreatment efficacy of SGLT2 inhibitor against ischemic stroke in non-diabetic mice using a very low dose, which had no effect on plasma glucose levels. Recent reports have also shown such beneficial effects of low-dose SGLT2 inhibitors, which are mediated by direct pharmacological action on each cell or organ and not by glycemic control. For instance, an experimental study demonstrated that administration of canagliflozin at a low dose improved albuminuria and mesangial expansion in *db/db* mice, in which tubular SGLT2-independent renoprotection was induced by direct inhibition of mesangial SGLT2[17]. They explained that canagliflozin inhibited protein kinase C activation and reactive oxygen species overproduction in mesangial cells. SGLT2 is also expressed in cultured retinal pericytes and regulates cell contraction via sodium-dependent glucose uptake and subsequent sodium-calcium exchanger (NCX)-mediated $Ca^{2+}$ influx and, thereby, may be involved in the regulation of retinal blood flow[18,19]. Taken together, the effects of low-dose luseogliflozin might not be primarily due to the inhibition of major SGLT2 function in renal proximal tubules, lowering plasma glucose levels, but possibly be mediated by direct action on SGLT2-expressing brain pericytes.

Pretreatment with luseogliflozin ameliorated infarct size and neurological dysfunction accompanied by an attenuation in pericyte loss. Because a high proportion of brain pericytes surround endothelial cells to maintain BBB integrity through close interactions, minimizing pericyte loss is a crucial strategy for preventing brain injury in ischemic stroke[20]. Our results suggest the possibility that SGLT2 inhibitors reduce pericyte loss and thus function as a BBB pre-conditioning drug. Another important possible mechanism for reducing ischemic injury is the improvement of brain microcirculation regulated by pericytes; however, it may not apparently affect macroscopic CBF within infarct areas, as evaluated by laser speckle flowmetry. Meanwhile, excessive uptake of glucose and $Na^+$ by upregulated SGLT2 may

induce the loss of pericytes through $Ca^{2+}$ overload. Further experiments are needed to evaluate intracellular $Ca^{2+}$ levels in pericytes after pMCAO.

Previous studies on glucose metabolism in brain pericytes, including the expression and function of glucose transporters, are limited. Although brain pericytes expressed various glucose transporters, including GLUTs, SGLT1, and SGLT2, our immunofluorescence images clearly showed that pericytes expressed high levels of SGLT2 in peri-infarct areas, suggesting that pericytes may actively absorb glucose via the transporter in an energy-depleted state. Although GLUTs play a role in facilitated transport[21], SGLT2 allows active glucose uptake using a sodium concentration gradient[22], indicating that SGLT2 may provide support for GLUTs for glucose absorption in glucose-deficient states. Brain pericytes may possibly use the acquired glucose not only for their own survival but also to share it with surrounding astrocytes. Indeed, a recent report revealed the metabolic cross-talk between pericytes and astrocytes by sharing glucose and mitochondria[23]. The pathophysiological reason of the upregulation of SGLT2 needs to be clarified in future.

While it is well known that GLUT1 expression is regulated by HIF-1α, the regulatory mechanisms for SGLT2 expression remain unclear. Previous studies showed that SGLT2 expression was upregulated in response to high glucose exposure in renal biopsies of patients with diabetic nephropathy[24] and also in mesangial cells in vitro[17]. Histone acetylation, accompanied by a high binding of hepatocyte nuclear factor 1a to the promoter region, was involved in *SLC5A2* upregulation in HK2 cells[25]. Meanwhile, another report suggested that ischemia-reperfusion reduced SGLT2 expression in HK2 cells[13], indicating an inverse correlation with HIF-1 induction. In contrast, our results showed significant upregulation of SGLT2 in response to OGD, a mimic of ischemia, which may induce hypoxic and metabolic stresses and upregulate various transcriptional factors. The differences in gene regulation may be due to varying cell properties and epigenetics among cell types and organs. Moreover, the immunofluorescence imagings suggest that the OGD may induce membrane trafficking of the SGLT2 protein. It has been reported that the post-translational modification of SGLT2 protein is regulated by membrane trafficking, which is dependent on extracellular glucose level[26]. Another report suggested that the activity of $Na^+$/glucose absorption from the apical membrane, a primary function of SGLT2, depends on the trafficking of the protein to the membrane[27]. Therefore, the activity of SGLT2 depends on not only the gene expression level but also membrane trafficking, i.e., a depletion of extracellular glucose may induce the membrane trafficking to activate glucose absorption.

AMPKα is recognized as an energy sensor and regulator activated by a decrease in intracellular ATP concentration,

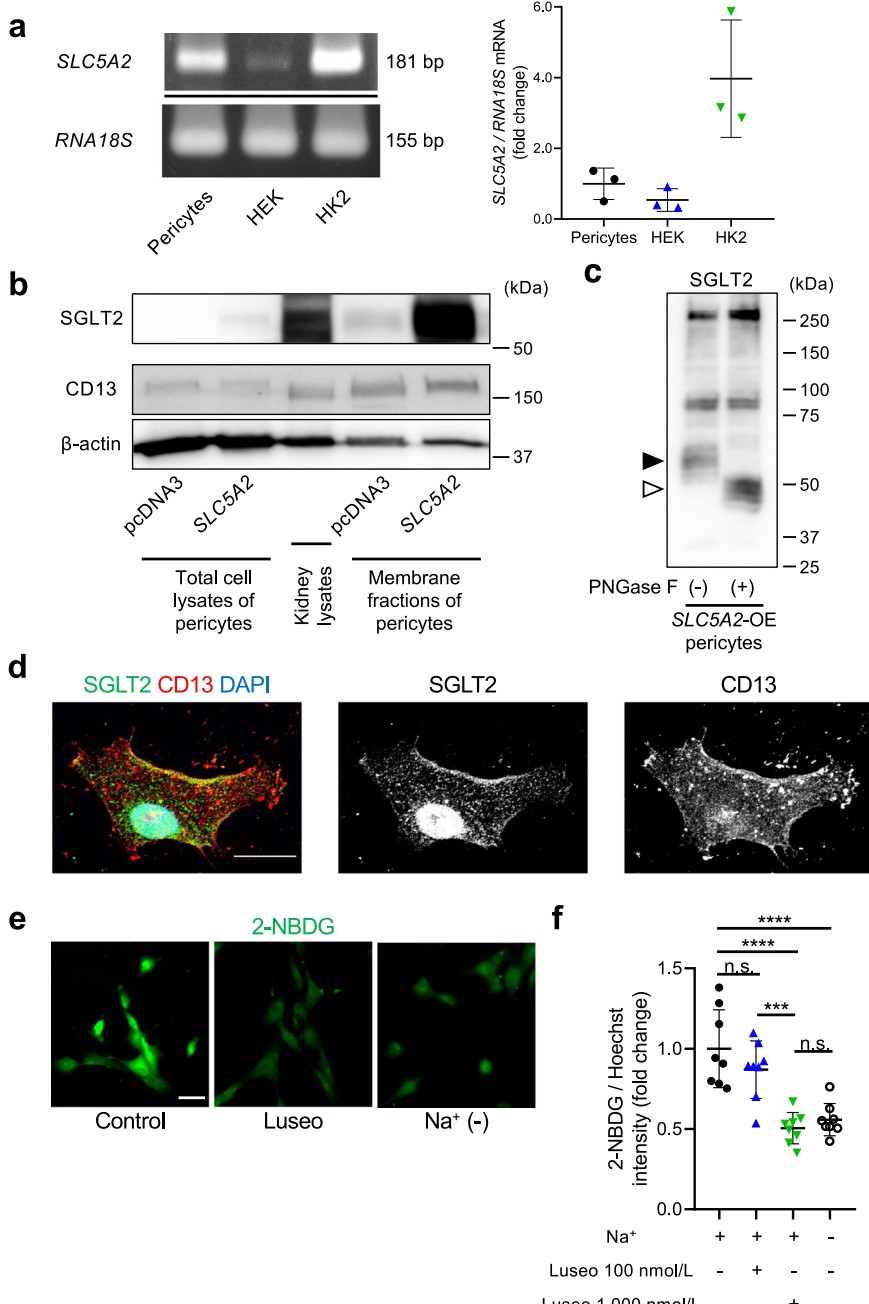

**Fig. 3 Glucose uptake in brain pericytes via SGLT2. a** Representative example of the expression of *SLC5A2* (SGLT2) in cultured human brain microvascular pericytes, HEK-293T, and HK2 cells assessed using RT-PCR and quantification using qPCR (*n* = 3). The mRNA expression was normalized to that of *RNA18S*. **b** Representative immunoblottings for SGLT2, CD13, and β-actin in cultured pericytes of total lysates or membrane fractions obtained by ultracentrifugation, which were transfected with pcDNA3 or pcDNA3-*SLC5A2*. Human kidney lysates were used as a positive control for SGLT2 expression. **c** A representative immunoblotting for SGLT2 in cultured brain pericytes after *SLC5A2* overexpression with or without PNGase F treatment. The black arrowhead indicates glycosylated SGLT2 and the white arrowhead indicates unglycosylated SGLT2. **d** Representative images of immunofluorescence for SGLT2 (green), CD13 (red), and DAPI in a cultured brain pericyte (scale bar = 20 μm). **e, f** (**e**) Brain pericytes were incubated in Na⁺-containing HEPES-buffered saline, Na⁺-free solution, or Na⁺-containing solution with 100 or 1,000 nmol/L luseogliflozin for 15 min, followed by incubation with additional 200 μmol/L of 2-NBDG (scale bar = 20 μm). **f** Fluorescence intensity of 2-NBDG was measured using a microplate reader and normalized to that of Hoechst 33342 (*n* = 8). Data are presented as dot-plots of individual experiments and mean values ± SD. n.s. (not significant) *P* > 0.05, \*\*\**P* < 0.001, \*\*\*\**P* < 0.0001 by One-way ANOVA followed by Bonferroni's post hoc test. SGLT2 sodium-glucose cotransporter 2, Luseo luseogliflozin, pMCAO permanent middle cerebral artery occlusion, 2-NBDG 2-deoxy-2-[(7-nitro-2,1,3-benzoxadiazol-4-yl) amino]-D-glucose.

which promotes various cytoprotective effects[28]. The following results suggest the mechanism by which luseogliflozin pre-administration attenuates pericyte loss in infarct lesions: (1) pericytes express SGLT2 in the brain, (2) luseogliflozin promotes mitochondrial biogenesis via the AMPKα pathway, conferring ischemic tolerance in cultured brain pericytes, and (3) luseogliflozin exhibits AMPKα activation in the mouse brain despite having no effect on blood glucose levels. Therefore, pre-administration of luseogliflozin may activate AMPKα and confer ischemic tolerance at least in brain pericytes. We should note

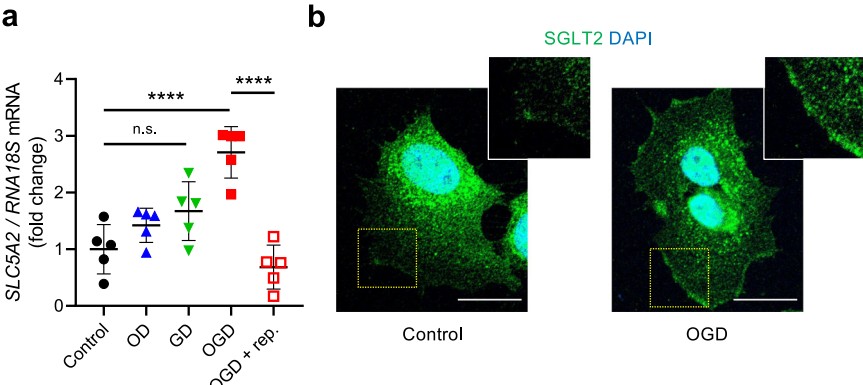

**Fig. 4 SGLT2 expression in cultured pericytes is upregulated by glucose and/or oxygen deprivation. a** Cultured brain pericytes were treated with GD for 16 h, OD for 16 h, OGD for 16 h, or OGD followed by 24-h reperfusion (rep.) with normal glucose and oxygen level. The mRNA expression level of *SLC5A2* was quantified using qPCR and normalized to that of *RNA18S* ($n = 5$). **b** Representative images of immunofluorescence for SGLT2 (green) and DAPI (blue) in pericytes treated with or without OGD for 16 h (scale bar = 20 μm). Data are presented as dot-plots of individual experiments and mean values ± SD. n.s. (not significant) $P > 0.05$, ****$P < 0.0001$ by One-way ANOVA followed by Bonferroni's post hoc test. SGLT2 sodium-glucose cotransporter 2, Luseo luseogliflozin, GD glucose deprivation, OD oxygen deprivation, OGD oxygen-glucose deprivation.

that this study did not demonstrate the pericyte-specific AMPKα activation and mitochondrial biogenesis in vivo. Although other neurovascular unit components may also be involved in AMPKα phosphorylation, the SGLT2 inhibitor may act directly on SGLT2-expressing pericytes to inhibit glucose uptake, decrease intracellular ATP concentration, and thereby induce AMPKα activation. Subsequently, treatment of cultured pericytes with SGLT2 inhibitor significantly upregulated a transcriptional co-activator PGC-1α, which coactivates the transcriptional function of NRFs on the TFAM promoter, a mitochondrial transcription factor regulating mtDNA transcription and replication[15,16]. The function of mtDNA is involved in a variety of diseases, including neurodegenerative diseases, cancer, aging, and BBB disruption[29–31]. Interestingly, the mechanism of action of metformin, a biguanide for type 2 diabetes, also involves the improvement of mitochondrial respiratory activity through AMPKα activity[32]. Similarly, ischemic tolerance with AMPKα phosphorylation has also been demonstrated with cerebral ischemic preconditioning[33]. Our results consistently demonstrated that pre-treatment with an SGLT2 inhibitor significantly reduced OGD-induced cell death of brain pericytes through acquired ischemic tolerance from increased mitochondrial activity.

This study has several limitations. First, we demonstrated the protective effect of luseogliflozin only in the acute phase of cerebral infarction by pre-treatment with the SGLT2 inhibitor for 2 weeks until pMCAO. Because pericytes promote functional recovery through tissue repair within infarct areas[34], it is necessary to determine the reason for the SGLT2 upregulation and the effect of long-term administration of an SGLT2 inhibitor on pericyte function during the chronic phase of ischemic stroke. Second, we cannot exclude the possibility that luseogliflozin did not act only on SGLT2 in brain pericytes but also on other glucose transporters, cells, and organs, including renal proximal tubules, although the selectivity for SGLT2 is high[35]. Thus, brain pericyte-specific deletion of *SLC5A2* will be helpful to demonstrate the direct action of the SGLT2 inhibitor in vivo. Third, we have not yet examined the effect of high or moderate doses of luseogliflozin on diabetic and non-diabetic mice. Before conducting clinical studies, additional experiments using non-diabetic and diabetic stroke animal models, as well as different dosages and time points of administration, are needed in the future. Fourth, the extent to which luseogliflozin passes through the BBB remains unknown. The concentration of luseogliflozin

used in vitro may not correspond to that in the pericellular microenvironment in vivo. Finally, we cannot judge the benefits and risks of SGLT2 inhibitors on the risk for stroke occurrence from this study. Further experiments are needed to address the direct action of SGLT2 inhibitors on brain pericytes and to clarify the potential utility of these drugs.

In conclusion, our results suggest that SGLT2 is expressed in brain pericytes and regulates glucose metabolism during ischemic stroke. Moreover, pre-administration of a low-dose SGLT2 inhibitor attenuates pericyte loss and BBB disruption by acquiring ischemic tolerance through AMPKα activation and mitochondrial biogenesis. Pre-stroke administration of low-dose SGLT2 inhibitors has the potential to become a novel preventive strategy to reduce brain damage and neurological dysfunction, even in non-diabetic patients.

## Methods

**Study animals.** All mice were maintained and handled according to the Guidelines for Proper Conduct of Animal Experiments by the National Science Council of Japan. The Animal Care and Use Review Committee of Kyushu University approved the animal experimental protocol (protocol number A20-107-1). C57BL/6JJcl mice were purchased from CLEA Japan (Tokyo, Japan). Male mice aged 8–11 weeks and weighing 20–35 g were used in the experiments. All experiments were reported according to the Animal Research: Reporting In Vivo Experiments (ARRIVE) guidelines. Sample size was calculated based on the previously published studies[34,36].

**Luseogliflozin administration.** Mice were randomly assigned into two groups: vehicle (no SGLT2 inhibitor) and experimental. The experimental group received the selective SGLT2 inhibitor, luseogliflozin, that was added to the CLEA Rodent Diet CE-2 powdered mouse chow (CLEA Japan) at a concentration of 0.0001%, which was sufficient to deliver an average daily dose of 0.1 mg/kg. We set the dose at a very low concentration to avoid any effect on the plasma glucose level, according to a previous study[37], in which 0.3–3.0 mg/kg luseogliflozin increased urinary glucose excretion and reduced non-fasting plasma glucose levels in *db/db* mice. Both groups had free access to food and water throughout the study period.

**Blood pressure and heart rate measurement.** Blood pressure and heart rate in mice were measured using a noninvasive tail-cuff BP system (BP-2000; Visitech Systems, Apex, NC, USA). Briefly, mice were acclimatized to the blood pressure monitoring procedure for 3 consecutive days prior to the experiment. Blood pressure and heart rate were measured immediately before and 3 days after pMCAO.

**Glucose tolerance test and urine glucose determination.** Casual plasma and urinary glucose levels were measured at approximately 10:00 a.m. Blood was collected from the tip of the tail (approximately 1 mm), and the glucose level was measured using FreeStyle Precision Neo (Abbott, Chicago, IL, USA). For the intraperitoneal glucose tolerance test, mice were injected with 1 g/kg of D-glucose

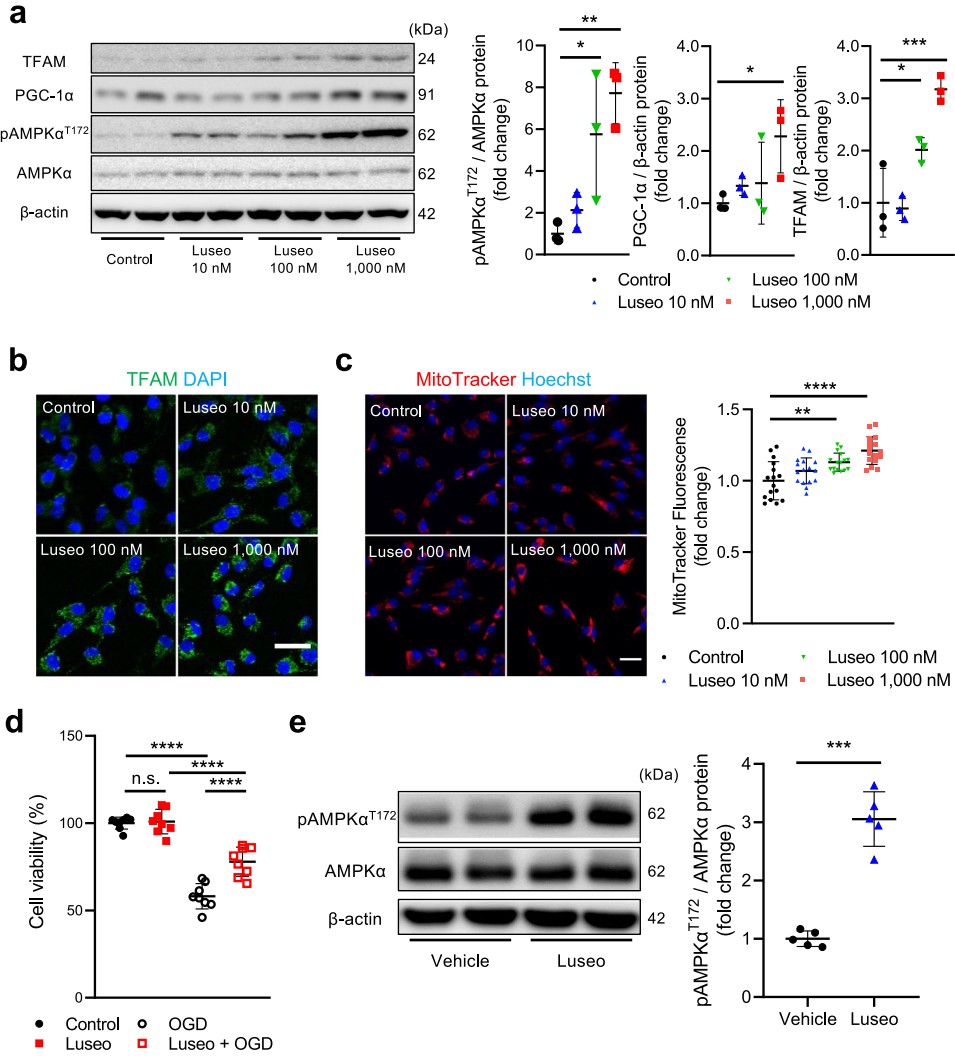

**Fig. 5 Luseogliflozin activates mitochondrial biogenesis by activating AMPKα and PGC-1α in brain pericytes. a–c** Cultured brain pericytes were treated with 0, 10, 100, or 1000 nmol/L luseogliflozin for 24 h. **a** Representative immunoblotting for pAMPKα$^{T172}$, AMPKα, PGC-1α, TFAM, and β-actin is shown, and their levels were quantified using densitometry. Protein expression level was normalized to that of AMPKα or β-actin ($n = 3$). **b** Representative images of immunofluorescence for TFAM (green) and DAPI (blue) in pericytes treated with luseogliflozin. **c** Representative live-cell imaging of cells labeled with MitoTracker (red) and Hoechst (blue) in pericytes treated with luseogliflozin and the quantification of the MitoTracker fluorescence intensity measured using a microplate reader ($n = 16$). **d** Cultured pericytes were treated with or without 1,000 nmol/L luseogliflozin for 24 h, followed by 16-h oxygen-glucose deprivation. Cell viability was evaluated using LDH assay ($n = 8$). **e** Luseogliflozin, mixed in food at a concentration of 0.0001%, was administered orally to C57BL/6JJcl mice for 2 weeks. Representative immunoblotting for pAMPKα$^{T172}$, AMPKα, and β-actin of the mice brain cortex is shown. Activated pAMPKα$^{T172}$ levels were quantified using densitometry and normalized to that of AMPKα ($n = 5$ mice). Data are presented as dot-plots of individual experiments and mean values ± standard deviation. n.s. (not significant) $P > 0.05$, *$P < 0.05$, **$P < 0.01$, ***$P < 0.001$, ****$P < 0.0001$ by One-way ANOVA followed by Bonferroni's post hoc test (**a, c, d**) or by unpaired $t$-test (**e**). Luseo luseogliflozin, TFAM mitochondrial transcription factor A, OGD oxygen-glucose deprivation.

(Wako Pure Chemical Industries, Osaka, Japan) intraperitoneally after an overnight fast. Blood glucose levels were measured before and 30, 60, and 120 min after the glucose injection.

**Middle cerebral artery occlusion model and exclusion criteria.** Focal cerebral ischemia was induced by pMCAO of the right middle cerebral artery (MCA) using a laser-induced photochemical reaction[34,36]. Mice were anesthetized by inhalation of 2% isoflurane and maintained under anesthesia with 1.5% isoflurane. Rectal temperatures were maintained at 35 °C–37 °C using a heating lamp. The right jugular vein was exposed, and a catheter was inserted into the superior vena cava for intravenous administration access. After the right distal MCA was exposed carefully, a diode-pumped solid-state laser was used to irradiate the distal MCA at a wavelength of 561 nm with an emitted power of 6 mW. Upon laser irradiation, a photosensitizing Rose Bengal dye solution (20 mg/kg) was administered intravenously for 90 s. After 4 min of irradiation, the laser beam was refocused on the MCA proximal to the first position, followed by another 4 min of irradiation. The right common carotid artery was ligated with a 6-0 silk suture.

We subjected a total of 129 male mice to the stroke model. To reduce the variability of infarction, we subjected only male mice to this study. Mice that prematurely died within 24 h after pMCAO ($n = 15$) and those with subcortical hemorrhage ($n = 8$) were excluded from the analysis.

**Measurement of CBF by laser speckle flowmetry.** Relative CBF was determined using laser speckle flowmetry (Omegazone OZ-2; Omegawave Inc., Tokyo, Japan), which generates high-resolution two-dimensional images showing a linear relationship with absolute CBF. Recordings were made through the skull under anesthesia with 1.5% isoflurane. The average CBF of the ischemic core located 2 mm posterior and 4 mm lateral to the bregma was measured in a region of interest (ROI; 900 pixels)[34].

**Assessment of blood-brain barrier breakdown.** The permeability of the BBB in the infarct areas was assessed by the leakage of Evans blue dye (Sigma-Aldrich #E2129). Briefly, the mice were transcardially perfused with ice-cold saline, and whole brains were isolated 180 min after intravenous injection of 2% solution of

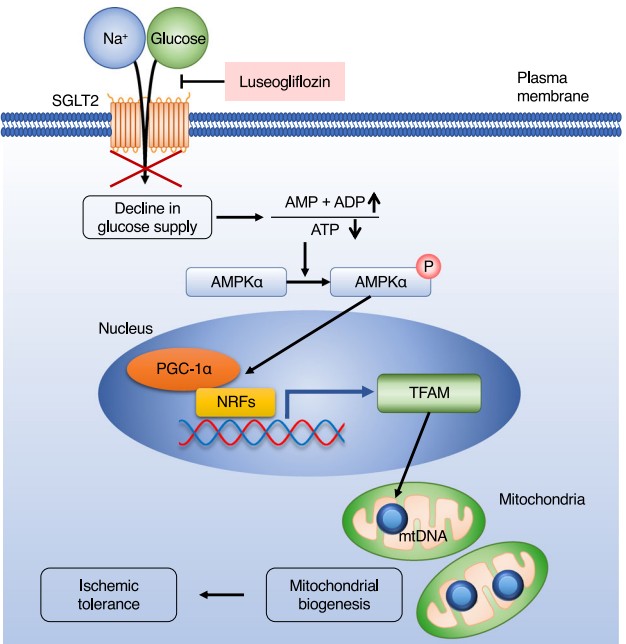

**Fig. 6 Schematic model depicting SGLT2 inhibitor-mediated acquisition of ischemic tolerance in brain pericytes.** Luseogliflozin, an SGLT2 inhibitor, inhibits Na+-dependent glucose uptake and activates AMPKα in response to depletion of intracellular glucose and ATP. Activated AMPKα enhances mitochondrial biogenesis via PGC-1α, NRF1, and NRF2 signaling, leading to ischemic tolerance acquisition. SGLT2 sodium-glucose cotransporter 2, TFAM mitochondrial transcription factor A.

Evans blue dye (4 mL/kg body weight). The brains were homogenized and immersed in 50% trichloroacetic acid to extract the dye. After incubation for 20 min at room temperature, the samples were centrifuged at 15,000 $g$ for 20 min. The optical density of the supernatant was measured at 650 nm using a microplate reader (ARVO X4; Perkin Elmer, Waltham, MA, USA).

**Behavioral tests.** All behavioral tests were performed in a blinded manner 3 days after the pMCAO. The rotarod test evaluated the balance and coordination of mice[38]. Prior to surgery, the mice were trained to balance on the rotating drum for 3 days, with 3 trials per day. After a 1-minute adaptation period on the rod at rest, the rod was continuously accelerated at 40 rpm for 10 min, and the length of time the mice remained on the rod (fall latency) was recorded. The neurological deficit score was evaluated using a six-point scale test with some modifications (Supplementary Table 3)[39].

**Microvessel isolation.** Brain microvessels were isolated from 8-week-old mice. Meninges were carefully removed from the forebrains. The gray matter was minced into approximately 1-mm³ pieces in ice-cold Hank's balanced salt solution (HBSS). Cell pellets containing the brain microvessels were separated by centrifugation in 20% bovine serum albumin-HBSS (2000 g, 10 min), and then filtered through 100-µm cell strainers and trapped using 40-µm cell strainers.

**Immunostaining.** Mice were euthanized by deep anesthesia using intraperitoneal administration of pentobarbital (75 mg/kg body weight) and were transcardially perfused with ice-cold saline and 4% paraformaldehyde (PFA; Wako Pure Chemical Industries). For paraffin sections, the brains were fixed with 4% PFA for 24 h. PFA-fixed 1 mm coronal slices were embedded in paraffin and cut into 4-µm thick sections using a microtome (Leica Biosystems). The sections were then deparaffinized, dehydrated through a graded series of ethanol solutions, washed in PBS, and subjected to epitope retrieval treatments. For frozen sections, the brains were fixed in 4% PFA for 1 h. PFA-fixed 3-mm coronal slices were embedded in an optimal cutting temperature compound and dissected into 10-µm thick sections using a cryostat (Leica Biosystems). After blocking with 5% skim milk for 30 min at room temperature, the sections were incubated with the following primary antibodies at 4 °C overnight: anti-microtubule-associated protein 2 (MAP2) (1:1,000; Sigma-Aldrich #M4403), anti-CD31 (1:200; BD Biosciences #550274), anti-CD31 (1:100; Abcam #ab56299), anti-CD13 (1:200; R&D Systems #AF2335), anti-PDGFRβ (1:100; R&D Systems #AF1042), and anti-SGLT2 (1:200; Abcam #ab85626). After thorough washing, the sections were incubated with appropriate secondary antibodies conjugated to Alexa Fluor dyes (Invitrogen, Thermo Fisher

Scientific, Waltham, MA, USA) or stained with 3,3-diaminobenzidine (DAB) using an appropriate kit (Nichirei, Tokyo, Japan). For DAB staining, endogenous peroxidase was inactivated with 0.3% hydrogen peroxide for 30 min before blocking with 5% skim milk. Nuclei were counterstained with DAPI. The sections were observed under a BIOREVO BZ-9000 microscope (Keyence Corporation, Osaka, Japan) or an A1R confocal microscope (Nikon Instruments, Melville, NY, USA). We used sections of the renal cortex of C57BL/6JJcl mice as a positive control of the SGLT2 expression (Supplementary Fig. 2a–b). To test the specificity of anti-SGLT2 antibody with the immunizing human SGLT2 peptide (Abcam #ab101414; the amino acid sequence was not available), the antibody was incubated overnight with the peptide (peptide/antibody molar ratio = 20:1) at 4 °C prior to immunohistochemistry and immunofluorescence use, as described above.

For the quantification of images, the objective fluorescent signal above the threshold was measured by a blinded investigator who was not involved in animal procedures using ImageJ software (National Institutes of Health). SGLT2-positive areas were quantified by assessing three random high-powered ROIs (200 × 200 µm) in peri- and intra-infarct areas at the bregma of each section. CD13- and CD31-positive areas were quantified by assessing five random high-powered fields (400 × 400 µm) in the infarct area at the bregma of each section. Infarct volumes were measured using a total of five sections spaced 1 mm apart (from bregma +2.0 mm to bregma –2.0 mm), as follows: Infarct volumes (%) = [(contralateral hemisphere) − (ipsilateral MAP2-positive areas)] × 100/ (contralateral hemisphere).

**Cell culture.** Human brain vascular pericytes, isolated from the normal brain, were purchased from ScienCell Research Laboratories (Carlsbad, CA, USA). Pericytes were cultured on poly-L-lysine (PLL)-coated dishes (Iwaki, Tokyo, Japan), using Pericyte Medium (ScienCell) containing 1% growth supplement, 2% fetal bovine serum, 100 U/mL penicillin, and 100 µg/mL streptomycin (Gibco, Thermo Fisher Scientific, Waltham, MA, USA) incubated at 37 °C in a humidified atmosphere with 5% CO₂. HEK-293T cells and HK-2 cells were obtained from American Type Culture Collection and used as controls.

**SLC5A2 overexpression.** The cDNA encoding human *SLC5A2* was purchased by from Addgene (plasmid #132241) and cloned into pcDNA3. The pcDNA3-FLAG-*SLC5A2* construct was verified by DNA sequencing using BigDye Terminator v1.1 Cycle Sequencing (ABI 3500xL DNA analyzer; Applied Biosystems, Waltham, MA, USA).

Lipofectin Transfection Reagent (Invitrogen, Thermo Fisher Scientific) or the Nucleofector Electroporation system (Amaxa Biosystems, Cologne, Germany) was used for the transfection of HEK-293T cells and pericytes, respectively, using the plasmid according to the manufacturer's protocol. After transfection, the cells were cultured for 48 h and then were used for the experiments.

**2-NBDG uptake measurement.** Glucose uptake was measured using a fluorescent substrate, 2-NBDG (Peptide Institute, Osaka, Japan). Cells were plated at a density of 10,000 cells/well in a 96-well plate and incubated in a culture medium for 24 h. The culture medium was then replaced with the following HEPES-buffered saline with 0, 100, or 1000 nmol/L luseogliflozin for 15 min: 140 mmol/L NaCl, 5 mmol/L KCl, 2.5 mmol/L CaCl₂, 1 mmol/L MgCl₂, 1 mmol/L KH₂PO₄, and 10 mmol/L HEPES (pH 7.4). In the Na+-free solution, NaCl was replaced with 140 mmol/L choline chloride. We added 200 µmol/L 2-NBDG and Hoechst 33342 incubated at 37 °C with 5% CO₂ for 30 min. After washing the cell plate, we measured 2-NBDG fluorescence with a filter for 485 nm excitation and 535 nm emission and Hoechst 33342 fluorescence with a filter for 355 nm excitation and 460 nm emission using a plate reader (ARVO X4; Perkin Elmer). To evaluate the 2-NBDG uptake per cell, the fluorescence intensity of 2-NBDG was normalized to that of Hoechst 33432.

**Oxygen and/or glucose deprivation, and low- or high-glucose stimulation.** Oxygen-glucose deprivation (OGD) or oxygen deprivation (OD) was performed using an airtight hypoxia chamber. Culture dishes were washed once with glucose-free DMEM (osmolality: 300–340 mOsm/kg; Gibco) and then subsequently filled with glucose-free DMEM (OGD) or 5 mmol/L glucose-containing DMEM (310–340 mOsm/kg) (OD) incubated at 37 °C in a humidified atmosphere with 5% CO₂, 1% O₂, and 94% N₂ for 16 h. Glucose deprivation (GD) was achieved using glucose-free DMEM incubated at 37 °C in a humidified atmosphere containing 5% CO₂ for 16 h. High glucose stimulation was performed on the cells maintained in DMEM with 2% FBS at 20 mmol/L glucose levels (320–355 mOsm/kg) incubated at 37 °C in a humidified atmosphere containing 5% CO₂ for 24, 96, and 168 h. The medium was changed every 24 h.

**Lactate dehydrogenase cytotoxic assay.** Cytotoxicity was determined by measuring the activity of lactate dehydrogenase (LDH) using the Cytotoxicity Detection KitPLUS (Roche Diagnostics GmbH, Mannheim, Germany) according to the manufacturer's instructions. Briefly, cells were seeded onto PLL-coated 96-well plates (Iwaki). After incubation with the reaction mixture for 30 min, the absorbance at 490 nm was measured using a microplate reader (ARVO X4; Perkin Elmer).

**RT-PCR**. Total RNA was prepared from cultured cells or brain tissues using the TRIzol reagent (Invitrogen, Thermo Fisher Scientific). Total RNA (1 µg) was reverse transcribed using a ReverTra Ace qPCR RT Kit (Toyobo, Osaka, Japan). Using the reverse transcription product as a template, PCR was performed using primers specific for the target genes. After preincubation at 94 °C for 5 min, PCR was performed with 35 cycles of denaturation at 94 °C for 30 s, annealing at 55 °C for 30 s, and elongation at 72 °C for 1 min.

Quantitative PCR was performed in duplicate using a LightCycler (Roche Diagnostics GmbH). Reaction volumes (20 µL) included 10 µL KOD SYBR qPCR mix (Toyobo), 0.5 µmol/L primers, and 2 µL cDNA. The mRNA copy numbers were normalized to 18 S ribosomal RNA (rRNA) as the internal control.

The following primers (Sigma-Aldrich, Tokyo, Japan) were used: mouse SGLT2 (*Slc5a2*; forward, 5′-GGTGCCTGGCTGGAAAGAATCTG-3′; reverse, 5′-TGCCA CCTCATCTGGGTAGAGAATG-3′), human SGLT2 (*SLC5A2*; forward, 5′-ACGC CTGATTCCCGAGTT-3′; reverse, 5′-GCGGTGGAGGTGCTTTCT-3′), human GLUT1 (*SLC2A1*; forward, 5′-CTTCACTGTCGTGTCGCTGT-3′; reverse, 5′-TGA AGAGTTCAGCCACGATG-3′), human GLUT2 (*SLC2A2*; forward, 5′-TGGTTTT CACTGCTGTCTCTG-3′; reverse, 5′-CATTCCAATTAGAAAGAGAGAACGTC-3′), human GLUT3 (*SLC2A3*; forward, 5′-CAATGCTCCTGAGAAGATCATAA-3′; reverse, 5′-AAAGCGGTTGACGAAGAGT-3′), human GLUT4 (*SLC2A4*; forward, 5′-CACAGTCTTCACCTTGGTCTCG-3′; reverse, 5′-GTAGCTCATGG CTGGAACTCG-3′), human SGLT1 (*SLC5A1*; forward, 5′-CCCTGGTTTTGGT GGTTGTG-3′; reverse, 5′-TCACCACCCCAGCCTTAATATAG-3′), human NRF1 (*NRF1*; forward, 5′- CCAGACGACGCAAGCATCAG-3′; reverse, 5′- GGGAT CTGGACCAGGCCATT-3′), human NRF2 (*NRF2*; forward, 5′-AAACCAGTG GATCTGCCAAC-3′; reverse, 5′- ACGTAGCCGAAGAAACCTCA -3′), and human/mouse 18 S rRNA (*RNA18S/Rna18s*; forward, 5′-AAACGGCTACCACA TCCAAG-3′; reverse, 5′-CCTCCAATGGATCCTCGTTA-3′).

**Immunoblotting**. Cultured cells and brain tissues were homogenized in M-PER Mammalian Protein Extraction Reagent (Thermo Fisher Scientific) for SGLT2 or in RIPA lysis buffer (50 mmol/L Tris-HCl, pH 7.5, 150 mmol/L NaCl, 1% NP-40, 0.5% deoxycholic acid, 0.1% SDS, 5 mmol/L EDTA, 10 mmol/L $Na_4P_2O_7$, 0.1 mmol/L $Na_3VO_4$, 1 mmol/L NaF, and protease inhibitor cocktail; Sigma-Aldrich) for the others. Protein concentration was determined using a BCA Protein Assay Kit (Pierce Biotechnology, Waltham, MA, USA). Reduced samples by 5% β-mercaptoethanol and SDS were subjected to SDS-PAGE (20 µg/lane) and then transferred onto PVDF membranes using Criterion Blotter tank (Bio-Rad Laboratories, Hercules, CA, USA) at 30 V for 90 min at 4 °C. The membranes were incubated for 60 min with ECL-advance blocking reagent (GE Healthcare, Chicago, IL, USA) at room temperature, and probed overnight at 4 °C with the primary antibodies, anti-SGLT2 (CST #14210), anti-CD13 (R&D Systems #AF2335), anti-β-actin (Sigma #A5441), anti-TFAM (CST #8076 S), anti-PGC-1α (Millipore #AB3242), anti-phosphorylated-AMPKα (Thr172) (pAMPKα$^{T172}$, CST #2535), anti-AMPKα (CST #5831), and anti-FLAG M2 (Sigma #F3165). Membranes were then washed and incubated with appropriate secondary antibodies (1:50,000 dilution, CST) for 60 min at room temperature. Blots were developed using ImmunoStar LD (Wako Pure Chemical Industries) according to the manufacturer's instructions.

To obtain membrane fractions, homogenized cell samples were centrifuged at 20,000 g for 10 min to pellet tissue debris, followed by centrifugation at 150,000 g for 60 min to remove cytosolic supernatant (Optima MAX-TL; Beckman Coulter, Brea, CA, USA).

Human kidney lysates (#PK-AB718-1345-N; PromoCell GmbH, Heidelberg, Germany) were used as positive control for SGLT2 expression. For glycosidase treatment, cell lysates were treated with PNGase-F (P0704S; New England Biolabs, Ipswich, MA, USA) at 37 °C for 60 min.

**Immunocytochemistry**. Cells or microvessels were fixed in 4% PFA, permeabilized with 0.1% Triton X-100 or methanol, and processed for immunostaining. The cells or microvessels were blocked for 30 min at room temperature with a blocking solution (3% BSA or 5% skim milk). The following primary antibodies were used: anti-SGLT2 (1:200; Abcam #ab85626), anti-CD31 (1:200; BD Biosciences #550274), anti-CD13 (1:200; R&D Systems #AF2335), and anti-TFAM (CST #8076). Nuclei were counterstained with DAPI or Hoechst 33342. An A1R confocal microscope was used for viewing (Nikon, Tokyo, Japan).

**Mitochondrial activity**. Quantitative PCR analysis was used to assess the mtDNA copy number in cells. Total intracellular DNA was isolated and purified using the NucleoSpin Tissue kit (Takara Bio, Shiga, Japan). Quantitative PCR was performed using the mitochondrial gene NADH dehydrogenase subunit 1 (*ND1*) to represent mtDNA and 18 S rRNA for nDNA. The following primers were used: *ND1* (forward, 5′-ATGGCCAACCTCCTACTCCT-3′; reverse: 5′-GCGGTGATGTAGA GGGTGAT-3′) and *RNA18S* (forward: 5′-CATTCGAACGTCTGCCCTATC-3′; reverse: 5′-CCTGCTGCCTTCCTTGGA-3′).

To visually detect active mitochondria in living cells, a low-toxicity cell-permeable fluorescent probe, MitoTracker Red CMX-Ros (Invitrogen, Thermo Fisher Scientific), was used to label high-potential mitochondria. The cells were probed with MitoTracker Red at a final concentration of 100 nmol/L for 15 min at 37 °C in the dark. Nuclei were counterstained with Hoechst 33342. MitoTracker

Red fluorescence was measured using a fluorescent plate reader with an excitation wavelength of 540 nm and an emission wavelength of 600 nm.

**Statistics and Reproducibility**. Data are expressed as mean ± standard deviation. Statistical analyses were performed using a two-sided unpaired Student's *t*-test or a One-way analysis of variance with post-hoc tests (Bonferroni's test). GraphPad Prism version 8.0 (GraphPad Software, San Diego, CA, USA) was used for all analyses. Statistical significance was set at $P < 0.05$.

**Reporting summary**. Further information on research design is available in the Nature Research Reporting Summary linked to this article.

## Data availability

Uncropped PCR gels and immunoblots are shown in Supplementary Fig. 6. The original data of the graphs and charts are shown in Supplementary Data 1. All other data provided in the article and supplementary files are available from the corresponding author upon reasonable request.

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

## Acknowledgements

We would like to thank Naoko Kasahara (Kyushu University) for providing technical support and Keiko Hirano (Kyushu University) for providing secretarial assistance. We appreciate the technical assistance from the Research Support Center, Research Center for Human Disease Modeling, Kyushu University Graduate School of Medical Sciences. We would also like to thank Editage (www.editage.com) for English language editing. This study was supported in part by Grants-in-Aid for Scientific Research (B 16H05439 and B 20H03791 to T. Kitazono and T.A.; C 20K09373 to T.A.; C 26462163, C 19K09530, and C 22K09209 to Y.W.; C 19K09511 and C 22K09236 to K.N.) and Grant-in-Aid for Research Activity Start-up (21K20693 to T.S.) from the Japan Society for the Promotion of Science (JSPS); a grant from Mochida Memorial Foundation for Medical and Pharmaceutical Research (K.N.); a grant from SENSHIN Medical Research Foundation, Japan (K.N., T.S., and T.A.); a grant from the Smoking Research Foundation (T.A.); and research grants from Astellas, Boehringer Ingelheim, Bristol-Myers Squibb, Daiichi Sankyo, Eisai, MSD, Sanofi, and Takeda (T. Kitazono and T.A.). This work was also supported by a research fund from Taisho Pharmaceutical Co., Ltd (Tokyo, Japan). The funders had no role in the study design, data collection, data analysis, or preparation of the manuscript. Luseogliflozin was obtained from Taisho Pharmaceutical Co., Ltd.

## Author contributions

M.T. designed and performed the experiments, analyzed the data, and wrote the manuscript. M.H., H.T., K.Y. and T.S. provided the general technical support. K.N., T. Kiyohara, Y.W., M.W., T.A. and T. Kitazono designed the experiments, provided general technical support, edited the manuscript, and directed the study.

## Competing interests

The authors declare no competing interests.
