## [Peer Review File · Communications Biology]

Reviewers' comments:

Reviewer #1 (Remarks to the Author):

[Comments]

The authors examined effects of pre-stroke inhibition of SGLT2 on ischemic tolerance in brain pericytes independent of the glucose-lowering effect and on ischemic brain injury. Pre-treatment with low-dose luseogliflozin significantly attenuated infarct volume, BBB disruption, and motor dysfunction after permanent MCA occlusion. Notably, luseogliflozin pre-treatment reduced pericytes loss in ischemic areas. They concluded that pre-stroke inhibition of SGLT2 induces ischemic tolerance in brain pericytes independent of the glucose-lowering effect, contributing to the attenuation of ischemic brain injury. These findings obtained by multiple methods are very interesting. However, the reviewer has some great concerns. This manuscript needs major revision.

<Major concerns>

- 1) In this study, the specificity of the antibody against SGLT2 (ab85626) is very important. The authors could detect a single band around 75 kDa in Western blotting using the antibody for SGLT2 (ab85626) (Fig. 3c). However, it has never been reported that a single band around 70-75 kDa in Western blotting using the antibody for SGLT2 (ab85626) can be detected. The authors should describe a special detailed method for Western blotting in Methods. In addition, the reviewer recommend the authors the immune-absorption test in Fig. 2, Fig. 3, and Fig. 4 in order to the specificity of the antibody, because the antigen is available. The reviewer would like to know if other bands of low molecular weight materials below 50 kDa were detected in Fig. 3c.
- 2) Judging from Fig. 2b~e and Fig. 4b, it seems that granular immunoreactivity of SGLT2 is localized in the cytoplasm of cells, suggesting localization of SGLT2 in intracellular organelles but not in cytoplasmic membrane. What do the authors think about granular immunoreactivity of SGLT2 in the cytoplasm of pericytes? What kinds of intracellular granular structures are immunopositive for SGLT2?
- 3) The authors reported that the uptake of 2-NBDG in Na⁺-containing buffered saline with 1000 nmol/L luseogliflozin was low, comparable to that in Na⁺-free buffered saline in cultures pericytes (Fig. 3e, f). How does luseogliflozin act on intracellular SGLT2?

Minor comments

- 1) In Supplemental Fig. 2a, an enlarged image showing proximal renal tubules is useful for confirming the specificity of the anti-SGLT2 antibody.

Reviewer #2 (Remarks to the Author):

In this interesting study, Takashima and colleagues examine the potential impact of low-dose SGLT2 inhibitor (luseogliflozin) on ischemic stroke in non-diabetic mice. They demonstrate that luseogliflozin reduces 2-NBDG uptake and ischemic stroke area, improving neurological readouts without altering blood glucose concentrations. My expertise is in metabolism, not neurobiology, so I will leave commenting on the neurobiology/functional analyses to reviewers with this expertise. Overall, I believe the work is carefully performed and represents a contribution to the field. My only concern is as follows:

I believe the novelty of this work is overstated. SGLT2 inhibitors have been studied in the setting of stroke (review/meta-analysis Tsai et al. Sci. Rep. 2021), including in non-diabetic subjects (reviewed by Li Cardiovasc. Endocrinol. Metab. 2019). I agree with the authors' point that further work to understand the potential utility of SGLT2i in non-diabetic subjects/independent of glucose lowering is important, but there should be a more balanced discussion of prior work.

POINT-BY-POINT RESPONSES

Reviewer #1

Overall comments: ...These findings obtained by multiple methods are very interesting. However, the reviewer has some great concerns. This manuscript needs major revision.

Response: We appreciate Reviewer #1 for the insightful comments on our study. We conducted additional experiments to confirm the specificity of anti-SGLT2 antibody and demonstrated the definite expression of SGLT2 on the cell surface membrane of pericytes. We believe that the additional information we present in the revised manuscript addresses Reviewer #1's concerns and substantiates our conclusions.

Comment 1: In this study, the specificity of the antibody against SGLT2 (ab85626) is very important. The authors could detect a single band around 75 kDa in Western blotting using the antibody for SGLT2 (ab85626) (Fig. 3c). However, it has never been reported that a single band around 70-75 kDa in Western blotting using the antibody for SGLT2 (ab85626) can be detected. The authors should describe a special detailed method for Western blotting in Methods. In addition, the reviewer recommends the authors the immune-absorption test in Fig. 2, Fig. 3, and Fig. 4 in order to the specificity of the antibody, because the antigen is available. The reviewer would like to know if other bands of low molecular weight materials below 50 kDa were detected in Fig. 3c.

Response: We appreciate Reviewer #1's insightful comments and helpful suggestions. According to the reviewer's suggestion, we conducted immunoblotting using SGLT2-overexpressing cells to confirm the specificity of the antibodies for SGLT2. The series of experiments revealed that the anti-SGLT2 antibody ab85626 (Abcam) was not appropriate for SGLT2 detection in immunoblotting analysis, but another antibody (CST #14210) was useful because SGLT2-overexpressing cells showed increased bands around 55–60 kDa. Therefore, we used CST #14210 in the subsequent immunoblotting and found the internal expression of SGLT2 in brain pericytes. We have revised figures and added the corresponding descriptions in page 12, lines 181–190.

1. We used HEK-293T cells with *SLC5A2* (SGLT2) overexpression to confirm the specificity of the antibody and found increased double bands around 55-60 kDa using anti-SGLT2 antibody #14210 (CST), which had almost the same size as FLAG-tag, indicating overexpressed SGLT2 protein

(**Supplementary Fig. S3b**). The size of the other bands were not increased by SGLT2 overexpression. Therefore, we considered that the 55–60-kDa band, detected by antibody #14210, was the specific band for SGLT2 protein. We used CST #14210 in the following experiments and found the internal and overexpressed SGLT2 in cultured brain pericytes, especially in the case of membrane fractions (**Fig. 3b-3c**)

2. Anti-SGLT2 antibody ab85626 (Abcam), which had been used previously, did not detect the overexpressed SGLT2 protein around 55–60 kDa in the immunoblotting analysis (data not shown). Thus, we judged that antibody ab85626, which detected multiple nonspecific bands, was inappropriate for immunoblotting analysis for SGLT2. Again, we thank the reviewer for the appropriate suggestion.
3. We used PNGase to examine the glycosylation of the protein because SGLT2, a membrane protein, is highly glycosylated in vivo. As shown in **Fig. 3c**, the 55–60-kDa band for SGLT2 shifted to approximately 50 kDa after treatment with PNGase F. This result was similar to that presented in the datasheet by Cell Signaling Technology.
4. We conducted immune-absorption tests to verify the specificity of anti-SGLT2 antibody ab85626 (Abcam) with the immunizing human SGLT2 peptide ab101414 in order to use the antibody for immunohistochemistry and immunofluorescence analyses. Revised **Supplementary Fig. S2a-S2b** shows the specific expression of SGLT2 in tubular cells of mouse kidney tissue and the treatment of the antibody with the corresponding peptide completely eliminated the fluorescent signal for SGLT2. Therefore, we judged that antibody ab85626 was appropriate for following immunohistochemistry and immunofluorescence analyses of mice tissue. Because the available peptide was limited, we performed the immune-absorption tests using only mouse kidney tissue, which expresses the highest SGLT2 in the body. We have added the corresponding descriptions in page 8, lines 135–136.
5. We have added the detailed methods for immunoblotting. In this revision, we arranged the protocol in several points to detect SGLT2, such as lysis buffer, sample reduction, wet tank system for blotting, and cooling during SDS-PAGE and blotting.

Comment 2: Judging from Fig. 2b~e and Fig. 4b, it seems that granular immunoreactivity of SGLT2 is localized in the cytoplasm of cells, suggesting localization of SGLT2 in intracellular organelles but not in cytoplasmic membrane. What do the authors think about granular immunoreactivity of SGLT2 in the cytoplasm of pericytes? What kinds of intracellular granular structures are immunopositive for SGLT2?

Response: As the reviewer accurately pointed out, the original immunofluorescence for SGLT2 appeared to be localized in the cytoplasm of pericytes. We adjusted the condition of experiments, and using both immunoblotting of membrane fractions and immunocytochemistry, we verified that SGLT2 was expressed not only in intracellular organelles but also in the cell surface membrane.

1. To improve the quality of immunofluorescence imaging and to elucidate the intracellular localization, we adjusted the condition of experiments, such as permeabilization and blocking solution. We have presented the immunostaining for SGLT2 using isolated microvessels in the new **Fig. 2d** and a cultured pericyte in the new **Fig. 3d**. We have also revised the immunofluorescence images in **Fig. 4b**. These results indicate that some amount of SGLT2 protein is located in the cell surface membrane of pericytes (particularly at the luminal side of microvessels), whereas others appeared to be located also in intracellular organelles.
2. We performed immunoblotting using membrane fractions of pericytes to clarify whether SGLT2 was expressed in cell surface membrane. As shown in the revised **Fig. 3b**, the membrane fractions of pericytes obtained by ultracentrifugation demonstrated the definite and increased expression of SGLT2 as well as cell surface protein CD13, compared with total cell lysate, indicating that SGLT2 is expressed in the cell surface.
3. The post-translational modification of SGLT2 protein is regulated by membrane trafficking into the plasma membrane, which is dependent on extracellular glucose level (Sunilkumar S. *Am J Physiol Cell Physiol* 2019). The primary function of SGLT2 is the absorption of glucose in a Na⁺-dependent manner from the apical membrane, and the activity of Na⁺/glucose absorption is reported to be dependent on the trafficking of the protein to the membrane (Katsurada K. *Circ Heart Fail.* 2021). In fact, our immunostaining may indicate the increased expression of SGLT2 in cell surface after OGD (**Fig. 4b**). Therefore, the activity of SGLT2 depends not only on the gene expression level but also on membrane trafficking. We considered that SGLT2 exists also in the intracellular organelles, but does not function usually, and that it would be effective after translocation to the cell surface membrane. We have added a discussion on intracellular localization and function of SGLT2 in page 23, lines 367–375.

Comment 3: *The authors reported that the uptake of 2-NBDG in Na⁺-containing buffered saline with 1000 nmol/L luseogliflozin was low, comparable to that in Na⁺-free buffered saline in cultures pericytes (Fig. 3e, f). How does luseogliflozin act on intracellular SGLT2?*

Response: We appreciate Reviewer #1's insightful comments. As we mentioned above, we consider

that the function of SGLT2 depends on the translocation to the cell surface membrane. Therefore, there is a high possibility that the Na⁺-dependent 2-NBDG uptake, which was inhibited by luseoglifrozoin, is mediated by SGLT2 that is translocated to cell surface membrane. We have mentioned this possible mechanism in page 23, lines 367–375.

Comment 4: In Supplemental Fig. 2a, an enlarged image showing proximal renal tubules is useful for confirming the specificity of the anti-SGLT2 antibody.

Response: According to Reviewer #1's helpful suggestion, we have added enlarged images of the immunohistochemistry and immunofluorescence analyses for SGLT2 in renal proximal tubules in **Supplementary Fig. S2a-S2b**. The revised figure indicates that the SGLT2 protein is highly expressed in proximal tubules, helping to confirm the specificity of the antibody.

Reviewer #2

Overall comments: I believe the work is carefully performed and represents a contribution to the field. :

Response: We appreciate Reviewer #2 for the positive comments on our study. We have revised our manuscript in response to Reviewer #2's comments regarding the novelty of our study compared with previous studies on SGLT2 and stroke.

Comments: I believe the novelty of this work is overstated. SGLT2 inhibitors have been studied in the setting of stroke (review/meta-analysis Tsai et al. Sci. Rep. 2021), including in non-diabetic subjects (reviewed by Li Cardiovasc. Endocrinol. Metab. 2019). I agree with the authors' point that further work to understand the potential utility of SGLT2i in non-diabetic subjects/independent of glucose lowering is important, but there should be a more balanced discussion of prior work.

Response: We thank the reviewer for the valuable comments. We agree with the reviewer's points that this study has several limitations in terms of the effect of SGLT2 inhibitor on stroke pathophysiology. Previous reviews, including the one Reviewer #2 kindly suggested, indicated that SGLT2 inhibitors has no significant effect on stroke development risk as determined by a meta-analysis of randomized controlled trials. For example, there was no significant difference in the occurrence of stroke between

placebo and empagliflozin groups in EMPA-REG study (Zinman B, *N Engl J Med.* 2015). The outcome of these clinical trials is the occurrence of stroke or cardiovascular death, i.e., the major adverse cardiovascular events (MACE). However, no clinical or experimental studies have evaluated the effect of pre-treatment with SGLT2 inhibitors on neurological or functional outcome after stroke. We have revised the manuscript to clarify the study background and limitations and avoided overstating the novelty and conclusion.

- We have added the description about previous clinical studies and clarified the objective of this study page 3, lines 48–51 and 52–54, as follows: “Although a recent meta-analysis demonstrated that SGLT2 inhibitors had a neutral effect on the risk of stroke development⁶, their benefits on functional or neurological outcomes after stroke have not yet been proven.” “Thus, it remains controversial whether SGLT2 inhibitors are beneficial or detrimental to the outcomes after ischemic stroke in both experimental and clinical studies.”
- We have omitted the expression “To the best of our knowledge, this is the first report” in page lines 306–308.
- We have clearly mentioned that we used the SGLT2 inhibitor before ischemic insult in page 3, line 54, page 4, line 76, page 20, line 298, and page 21, lines 318–319. Moreover, we have changed the expression “BBB protective drug” to “BBB pre-conditioning drug” in page 22, line 339.
- We have added a limitation as follows: “Finally, we cannot judge the benefits and risks of SGLT2 inhibitors on the risk for stroke occurrence from this study. Further experiments are needed to address the direct action of SGLT2 inhibitors on brain pericytes and to clarify the potential utility of these drugs.” in page 25, lines 415–418.
- We have changed the conclusion from “to become a novel therapeutic strategy” to “to become a novel preventive strategy to reduce brain damage and neurological dysfunction” in page 25, lines 422–424.

REVIEWERS' COMMENTS:

Reviewer #1 (Remarks to the Author):

The authors appropriately responded to the reviewer's comments and revised the manuscript. The revised manuscript has been improved.

Reviewer #2 (Remarks to the Author):

I congratulate the authors on their interesting and well-performed work.